# The Role of Phosphatidylinositol 3-Kinase Catalytic Subunit Type 3 in the Pathogenesis of Human Cancer

**DOI:** 10.3390/ijms222010964

**Published:** 2021-10-11

**Authors:** Chien-An Chu, Yi-Wen Wang, Yi-Lin Chen, Hui-Wen Chen, Jing-Jing Chuang, Hong-Yi Chang, Chung-Liang Ho, Chen Chang, Nan-Haw Chow, Chung-Ta Lee

**Affiliations:** 1Department of Pathology, College of Medicine, National Cheng Kung University, Tainan 701401, Taiwan; s58001129@gmail.com (C.-A.C.); ywwang0228@gmail.com (Y.-W.W.); clh9@mail.ncku.edu.tw (C.-L.H.); cornea100@hotmail.com (C.C.); 2Department of Pathology, National Cheng Kung University Hospital, Tainan 704302, Taiwan; emerald@mail.ncku.edu.tw (Y.-L.C.); fairytalehome75@gmail.com (H.-W.C.); 3Department of Microbiology, Immunology and Biopharmaceuticals, National Chiayi University, Chiayi 600355, Taiwan; yihyeh@mail.ncyu.edu.tw; 4Department of Biotechnology and Food Technology, College of Engineering, Southern Taiwan University of Science and Technology, Tainan 710301, Taiwan; czeus1974@gmail.com; 5Institute of Molecular Medicine, College of Medicine, National Cheng Kung University, Tainan 701401, Taiwan; 6Institute of Basic Medical Sciences, College of Medicine, National Cheng Kung University, Tainan 701401, Taiwan; 7Department of Pathology, National Cheng Kung University Hospital, Dou-Liou Branch, Yunlin, Tainan 640003, Taiwan

**Keywords:** PIK3C3, Vps34, autophagy, colorectal cancer, cancer

## Abstract

Phosphatidylinositol 3-kinase catalytic subunit type 3 (PIK3C3), the mammalian ortholog of yeast vesicular protein sorting 34 (Vps34), belongs to the phosphoinositide 3-kinase (PI3K) family. PIK3C3 can phosphorylate phosphatidylinositol (PtdIns) to generate phosphatidylinositol 3-phosphate (PI3P), a phospholipid central to autophagy. Inhibition of PIK3C3 successfully inhibits autophagy. Autophagy maintains cell survival when modifications occur in the cellular environment and helps tumor cells resist metabolic stress and cancer treatment. In addition, PIK3C3 could induce oncogenic transformation and enhance tumor cell proliferation, growth, and invasion through mechanisms independent of autophagy. This review addresses the structural and functional features, tissue distribution, and expression pattern of PIK3C3 in a variety of human tumors and highlights the underlying mechanisms involved in carcinogenesis. The implications in cancer biology, patient prognosis prediction, and cancer therapy are discussed. Altogether, the discovery of pharmacological inhibitors of PIK3C3 could reveal novel strategies for improving treatment outcomes for PIK3C3-mediated human diseases.

## 1. Introduction

Autophagy is an adaptive cellular response to stress that involves the formation of autophagosomes. Microtubule-associated protein 1 light chain 3 II (LC3 II, yeast Apg8/Aut7/Cvt5/Atg8) plays an important role in autophagy and is used as a biomarker of this process. Autophagosome formation depends on the initiation and nucleation of membrane phagophores and elongation [1]. Autophagosomes engulf cytosolic organelles and proteins and fuse with lysosomes to recycle macromolecules in preparation for energy production [2]. Autophagy is involved in the pathogenesis of human diseases. In terms of epithelial cancer, autophagy has multifaceted roles of suppressing tumorigenesis or promoting carcinogenesis through different mechanisms [3,4,5].

Phosphatidylinositol 3-kinase catalytic subunit type 3/class III (PIK3C3/Vps34 in yeast) is an important autophagy-related protein. The PIK3C3 gene encodes an 887-amino acid lipid kinase [6]. PIK3C3 induces autophagosome nucleation through the formation of PIK3C3 complex 1 (PIK3C3-C1) by phosphorylation of the 3-OH of phosphatidylinositol to generate phosphatidylinositol 3-phosphate (PI3P) [7,8]. PIK3C3-C1 includes PIK3C3, phosphoinositide-3-kinase regulatory subunit 4 (PIK3R4, Vps15/p150), Beclin1 (Vps30/ATG6/BECN1), autophagy-related 14 (Atg14, BARKOR), nuclear receptor binding factor 2 (NRBF2, ATG38), and activating molecule in BECN1-regulated autophagy (AMBRA1, CRL4). PIK3C3 can also induce autophagy and endocytic trafficking through the formation of PIK3C3 complex 2 (PIK3C3-C2). The complex includes PIK3C3, PIK3R4, BECN1, and UV radiation resistance associated gene (UVRAG) proteins [8,9] (Figure 1). PIK3C3 plays a key role in many human cancers, including hepatocellular carcinoma (HCC) [10], lung cancer [11], colorectal cancer (CRC) [12], and breast cancer [13], but the underlying mechanisms involved in tumorigenesis remain elusive. Moreover, PIK3C3 has been suggested to play a role in nonneoplastic diseases, such as autoimmune diseases, Alzheimer’s disease, fibroproliferative liver diseases, and muscular dystrophy [14,15,16,17,18,19].

In this review, the characteristics of PIK3C3 will be summarized with an emphasis on its structure, biological functions, tissue distribution, and expression patterns in the human body. More details will be provided on the mechanisms of PIK3C3 in carcinogenesis. Then, its implications in tumor biology and patient prognosis prediction and its application in cancer therapy will be discussed.

## 2. Phosphoinositide 3-Kinases (PI3Ks): Gene, Protein, and Structure

### 2.1. Structure Domain

PI3Ks include class I PI3Ks (PIK3C1), class II PI3Ks (PIK3C2), and PIK3C3. PIK3C1 inhibits autophagy through phosphorylation of the 3-position of the inositol headgroup in phosphatidylinositol (PtdIns) 4,5-bisphosphate in the regulation of the AKT-mTOR pathway. However, PIK3C2 and PIK3C3 mediate the initiation of autophagy by phosphorylating PtdIns to PI3P [20,21]. The core catalytic structure of the three classes of PI3Ks is the C2 domain, a helical domain and a PI3K catalytic domain [21].

The structure domain of PIK3C1 includes the p85 binding domain, RAS binding domain (RBs of D), C2 domain, helical domain and PI3K catalytic domain. PIK3C1 contains IA (three distinct catalytic subunits of p110α, p110β, and p110δ) and IB (p110γ catalytic subunit) subclasses, and PIK3C1 is composed of dimers of one catalytic and regulatory subunit (p85, p50, p55, p84, p87 or p101) [21]. Compared with class I PI3Ks, PIK3C2 has a clathrin binding domain, TACC3-binding domain, or PX domain [22]. In contrast, PIK3C3 has three domains: the C2, helical, and PI3K catalytic domains [23] (Figure 1).

### 2.2. PIK3C2

The catalytic subunit of PIK3C2 has a Ras-binding domain and PI3K catalytic domain but lacks a regulatory subunit [23]. PIK3C2 is phosphorylated at the 3-position of the inositol ring of selected phosphoinositides (creating PI3P and phosphatidylinositol 3,4-bisphosphate (PtdIns(3,4)P2)) and is composed of PIK3C2A (α), PIK3C2B (β) and PIK3C2G (γ) [24]. The N-terminal extension of PIK3C2A, which has specific roles in the activation of enzymes, possesses a clathrin-binding domain and TACC3-binding domain. PIK3CA can mediate Clathrin and TACC and regulate breast cancer cell mitosis [25]. In a different way, the N-terminal extension of PIK3C2A and PIK3C2B possesses clathrin binding properties [22,24,26] (Figure 1). PIK3C2A and B can regulate endosomal trafficking and cell migration through mediating clathrin [22,27].

PIK3C2A is involved in vesicular trafficking by producing PI3P in the regulation of Rab8 and Rab11 [22,28,29]. Furthermore, PIK3C2A regulates autophagosome biogenesis by PI(3)P-mediated pathway [22]. In breast cancer models, PIK3C2A is required for genomic stability by regulating mitotic spindle formation. The reduction of PIK3C2A in breast cancer models initially impairs tumor growth but later leads to the convergent evolution of fast-growing clones with mitotic checkpoint defects [25]. Mutation of PIK3C2A results in syndromic short stature, skeletal abnormalities, and cataracts by mediating ciliary dysfunction [30]. PIK3C2A and PIK3C2B were suggested to play a role in tumor progression [31]. In human lung carcinoma cells, the increased expression of PIK3C2A and B is associated with talc particle-induced cell death and IL-6 secretion. These two effects are key cellular events leading to fibrosis [32]. Moreover, downregulation of PIK3C2B delays cell division and potentiates the effect of docetaxel on cancer cell growth [33]. In CRC, PIK3C2B activation by upregulated IQGAP3 can promote tumor growth and metastasis [34]. PIK3C2G produces PI3P and is involved in insulin/AKT2 and endosomal signaling pathways. It also regulates glycogen synthase by producing PI(3,4)P2 [31]. PIK3C2G could be a predictor for recurrence and overall survival of stage III CRC patients, especially after oxaliplatin-based therapy [35]. PIK3C2G could also be a prognostic biomarker in many types of cancer [22]. Altogether, the three types of PIK3C2 play different roles in human diseases, especially in epithelial carcinogenesis.

### 2.3. PIK3C3

#### 2.3.1. PIK3C3-C1

PIK3C3-C1 or PIK3C3-C2 is formed by the combination of different regulatory subunits [9]. PIK3C3-C1, which is comprised of the PIK3C3 catalytic subunit and regulator subunits of PIK3R4, BECN1, Atg14, NRBF2, and AMBRA1, is essential for autophagy initiation [8,9,36] (Figure 1). Regarding the structure of PIK3C3-C1 (yeast and human), PIK3C3 and PIK3CR4 belong to the catalytic arm [37,38,39]. The aromatic finger motifs of the Bara domain in BECN1 and the BATS domain (which does not exist in yeast) of ATG14 are adaptor arms, and the functions of these domains are related to kinase activity and the ability to bind to the membrane [40,41,42,43]. PIK3R4 is a serine/threonine protein kinase. In the PIK3C3 complex, PIK3R4 regulates the stability and activity of the catalytic domain [44]. In PIK3C3-C1, the coiled-coil domain (CCD) of BECN1 can bind to ATG14 and PIK3C3, and the ECD binds to PIK3C3 [42,45]. NRBF2 contributes to PtdIns3P synthesis with PIK3C3. The CCD of NRBF2 is responsible for membrane binding and microtubule interactions, and the trafficking molecule (MIT) domain of NRBF2 binds to PIK3R4, BECN1 and ATG14 [8]. AMBRA1 induces autophagy by interacting with BECN1 of PIK3C3-C1 in the initiation stage of autophagy [9]. The Ser-R domain of AMBRA1 can bind to BECN1 [46]. In the termination of autophagy (autophagolysosome degradation), the degradation of AM-BRA1 is induced by E3 ligase ring finger protein 2 (RNF2) [46,47]. In contrast, the protein abundance of the proautophagic factor AMBRA1 is limited by Cullin-4 in the presence of nutrients during autophagy termination [46,48]. Another BECN1 regulator, DIRAS3, induces autophagy initiation by binding with BECN1 and PIK3C3 in PIK3C3-C1 to prevent autophagy attenuation by the BECN1–BCL2 interaction [49] (Figure 2).

Several studies have demonstrated that BECN1 and ATG14 may be involved in cancer progression or chemoresistance. BECN1 is one of the major regulators of the PIK3C3 complex in autophagosome nucleation and endocytic trafficking. Silvia et al. discovered BECN1 mutations in 215 out of 38,262 unique samples (0.5%) in the COSMIC database [50]. Bcl-2, an important autophagy inhibitor, inhibits autophagy initiation by binding to the tumor suppressor BECN1 [50,51]. Loss of BECN1 expression was demonstrated in ovarian, breast and prostate carcinomas and is related to patient survival in breast and stomach carcinomas [52,53,54,55,56,57,58,59]. In connection with BECN1, oncogenic tyrosine kinases, such as EGFR and HER2, also regulate autophagy [50]. In the MG-63 osteosarcoma cell line, rapamycin inhibits proliferation and promotes apoptosis in association with increased protein levels of PIK3C3 and BECN1 [60]. In a CRC cell model, RNF216 induced cell proliferation and migration by inhibiting BECN1-mediated autophagy [61]. ULK1, which is an activator of BECN1, induces autophagy by phosphorylating BECN1 and activating VPS34 lipid kinase [62]. Activation of the PIK3C3 complex by DAPK3, which phosphorylates ULK1 at Ser556, suppresses migration, invasion, and tumor growth in gastric cancer in vitro and is related to favorable patient survival outcomes [63] (Figure 2). Conversely, enhanced autophagosome formation mediated by PTP4A3, an activator of PIK3C3-BECN1 complex, may promote the proliferation of ovarian cancer cells [64]. In addition, reduced BECN1/Vps15/Vps34 complex activity mediated by the interaction of BECN1 with androgen receptor can inhibit ERK-mediated growth factor signaling and suppress the growth of castration-resistant prostate cancer [58]. Taken together, these findings indicate that the regulation of the PIK3C3 complex by BECN1 could stimulate or inhibit tumors in different microenvironments.

ATG14, another regulatory subunit of PIK3C3-C1, takes part in autophagosome nucleation and promotes the fusion of autophagosomes and lysosomes [65]. ATG14 could alter the sensitivity to targeted drugs (gemcitabine, cisplatin, and sorafenib) by affecting the expression of microRNAs (miRNAs) in pancreatic, ovarian, colorectal, and liver cancers [65,66,67,68]. The PIK3C3-C1 regulator AMPRA1 acts as a tumor suppressor through the regulation of D-type cyclins [69,70]. The loss of AMPRA1 promotes the growth and invasion of melanoma [71].

#### 2.3.2. PIK3C3-C2

The major role of PIK3C3-C2 is to regulate vesicle trafficking and autophagosome formation. The structure of PIK3C3-C2 is similar to that of PIK3C3-C1, except that PIK3C3-C2 replaces ATG14 with UVRAG (Vps38) [9]. UVRAG and BECN1 form a regulatory subcomplex through parallel coiled-coil interactions and assemble with the catalytic subcomplex (PIK3C3 and PIK3CR4) to produce PIK3C3-C2 [41]. In PIK3C3-C2, the end of BECN11 and UVRAG bind to the membrane through their BARA (BECN1) and BARA2 (UVRAG) domains [41]. Rubicon can block PIK3C3-C2-induced autophagy and endosomal trafficking by binding to the BARA domain of BECN1 [41,50] (Figure 2). UVRAG can regulate autophagosome maturation through a PIK3C3 complex-independent pathway, which interacts with the class C VPS complex (C-VPS) and activates Rab7a [72].

UVRAG may play diverse roles as an oncogene or tumor suppressor in different types or environments of carcinoma. UVRAG-mediated activation of the BECN1-PI3KC3 complex induces autophagy and suppresses the proliferation and tumorigenicity of human colon cancer cells [73]. However, it induces the proliferation, migration, and invasion of breast cancer through CDK and integrin beta/Src signaling [74]. In HCC, the ubiquitination of UVRAG by SMURF1 induces autophagy maturation and inhibits tumor growth [75]. In a mouse model, UVRAG was found to drive inflammation and tumorigenesis in association with autophagy [76].

UVRAG has been reported to affect radiosensitivity and chemosensitivity. Suppressing UVRAG induces radiosensitivity in hypopharyngeal SCC and pancreatic cancer cells [77,78]. In CRC, the induction of UVRAG by HDAC1 promotes the chemoresistance of HCT116 cells to 5-fluorouracil (5-FU) [79].

## 3. Expression of PIK3C3 in Human Tissues

Analysis of The Human Protein Atlas (https://www.proteinatlas.org/ (accessed on 3 July 2021)) showed that PIK3C3 protein is ubiquitously expressed in human organs, with high levels observed in the adrenal gland, nasopharynx, oral mucosa, esophagus, stomach, duodenum, small intestine, colon, rectum, appendix, liver, gallbladder, ovary, cervix, uterine, placenta, muscle, tissues, skin, tonsil, and bone marrow [80,81].

A deep proteome and transcriptome analysis of 29 healthy human tissues in the EMBL-EBI Gene Expression Atlas database (https://www.ebi.ac.uk/gxa/home (accessed on 6 July 2021)) showed that the expression of PIK3C3 protein is relatively high in the human ovary, adrenal gland, colon, heart, urinary bladder, lung, smell intestine, spleen, stomach, and vermiform appendix [82]. The reanalysis of the Pandey lab’s draft of the human proteome in the EMBL-EBI database showed that the expression of PIK3C3 protein is relatively high in human testes, rectum, and ovary tissues and in monocytes and CD4-positive T cells.

At the transcriptomic level, PIK3C3 expression is high in the human cerebellum, spinal cord, occipital cortex, caudate nucleus, amygdala, dorsal thalamus, middle temporal gyrus, substantia nigra, parietal lope, pituitary gland, and middle frontal gyrus (RNA-Seq Cap Analysis of Gene Expression (CAGE) of human tissues in the RIKEN FANTOM5 project). In the RNA-seq analysis of 53 human tissue samples from the Genotype-Tissue Expression (GTEx) Project of the EMBL-EBI database, PIK3C3 RNA expression was high in human EBV-transformed lymphocytes, cerebellum tissue, uterine tissue, transformed skin fibroblasts, cerebellar hemisphere tissue, the tibial artery, and the tibial nerve.

## 4. Regulatory Factors, Pathways and Interaction Molecules for PIK3C3

The activity of PIK3C3 is tightly regulated by acetylation, transcription, posttranslational ubiquitination, modification, and miRNA regulation. PIK3C3 can be repressed by EP300/p300-mediated acetylation at the K711 and K29 sites, which limits the binding of VPS34 to its substrate PtdIns and blocks VPS34-Beclin 1 complex formation [83,84]. The kinase function of PIK3C3 is directly regulated by AMPK, which phosphorylates the Thr163 and Ser165 sites of PIK3C3 and is transcriptionally regulated through AMPK phosphorylation of the Ser555, Ser558 and Ser626 sites of the transcription factor FOXO3 [85,86]. STAT3, which forms a dimer by being phosphorylated by SRC or JAK kinase on Tyr705 and enters the nucleus, can indirectly regulate the expression of PIK3C3 protein [87]. Increased Y705-STAT3Y705 phosphorylation is associated with the downregulation of PIK3C3 [88]. PIK3C3 is inhibited by K48-linked ubiquitination of PIK3C3, which can be suppressed by the E3 ubiquitin ligase NEDD4/NEDD4-1 recruiting USP13 [89]. Induction of autophagy through assembly of the Beclin-1-Vps34-Atg14 complex can be activated by PIK3C3 SUMOylation by SUMO1 [90,91]. PIK3C3 is also regulated by transcriptional miRNA degradation. Both miR-338-5p and miR-340-5p can target the 3’ untranslated region (UTR) of PIK3C3, resulting in suppressed RNA expression [12,92]. MiR-143, which is regulated by GRP78, can inhibit the expression of PIK3C3 RNA and protein [93] (Figure 2).

PIK3C3 regulates autophagy-related pathways and extracellular vesicle trafficking-related pathways through the formation of complex I or complex II of PIK3C3 and the phosphorylation of PtdIns to PtdIns3P (PI3P) [9] (Figure 2). Phospholipid PI3P can be detected not only in autophagosomes and endosomes, but also in multivesicular bodies, midbodies, peroxisomes, and omegasomes [94]. In mammals, PI3P regulates vesicle trafficking by binding with the FYVE domain on early endosomes, spherical multivesicular bodies, and the nucleus [95]. In the endoplasmic reticulum (ER), mitochondria, and Golgi apparatus, PI3P induces the initiation of autophagy by migrating, forming puncta, and then forming an omegasome (Ring sharp) [96,97]. The PtdIns3P-positive compartment is on the outside, while LC3-positive autophagosomal membranes are on the inside of the omegasome. Then, a ring shape forms a double membrane and becomes the initiated type of autophagosome [96]. In the late stage of autophagy, PI3P and the ATG12–ATG5 complex-related protein tectonin beta-propeller repeat containing 1 (TECPR1) regulate the fusion of lysosomes with autophagosomes. Furthermore, FYVE and coiled-coil domain containing 1 (FYCO1) are also involved in the maturation and trafficking of autophagolysosomes [97,98,99]. In the termination of autophagy, cullin-3 and Kelch-like protein 20 (KLHL20) bind to BECN1 and PIK3C3 and ubiquitinate them for proteasomal degradation [46,100].

## 5. Biological Functions of PIK3C3

### 5.1. Proliferation, Apoptosis, and Tumor Growth

PIK3C3 is essential for cell proliferation [6] (Table 1). Jia et al. found that PIK3C3-deficient T lymphocytes cannot proliferate efficiently [101]. Induction of PIK3C3-C1 by SUMOylation of SUMO1 can promote the proliferation of vascular smooth muscle cells under nonhypoxic and hypoxic conditions [90]. Knocking down PIK3C3 in mouse embryonal fibroblasts suppressed tumor growth in a xenograft model of tuberous sclerosis complex [102].

PIK3C3 promotes proliferation and tumor growth in various cancer models. In RKO colon cancer cells, PIK3C3 inhibition using shRNA or PIK-III (a PIK3C3 inhibitor) enhanced lysosomal delivery and degradation of the transferrin receptor, leading to persistent cellular iron deprivation and impaired cell growth. This highlights the key role of PIK3C3 in regulating cell growth through iron metabolism [103]. PIK3C3 inhibition using siRNA was shown to enhance apoptotic signaling and drive HCT116 cells to undergo apoptosis [104]. In CT26 colon cancer cells and B16-F10 melanoma cells, targeting PIK3C3 with siRNA or PIK3C3 inhibitors reduced autophagy and tumor growth and improved mouse survival by inducing the infiltration of natural killer, CD8+, and CD4+ T effector cells [105] (Figure 3).

PIK3C3 is a mediator of EGFR nuclear trafficking, which has been shown to inhibit the transcription of the tumor suppressor gene ARF [111,115,116]. PIK3C3 facilitates EGFR nuclear transport and binding to the ARF promoter and inhibits ARF transcription [111]. Inhibition of PIK3C3 using siRNA restrains nuclear EGFR localization and restores ARF expression, leading to the apoptosis of H1975 lung cancer cells [111,115].

In an MCF-7 breast cancer cell model, PIK3C3 was found to stimulate ERK pathway-related tumor progression through protein kinase C-δ (PKC-δ)-mediated activation of p62 [108] (Figure 2). PIK3C3 induces the PKC-δ-dependent phosphorylation of p62, which leads to positive feedback on the Nrf2-dependent transcription of oncogenes and activation of the ERK pathway. Overexpression of PIK3C3 promotes the colony formation of MCF-7 cells in vitro and tumor growth in vivo [108].

In a study using the ETV6-RUNX1-positive leukemia cell lines REH and REHS1, the induction of PIK3C3 was transcriptionally regulated by ETV6-RUNX1 and correlated with high levels of autophagy [114]. Knockdown of PIK3C3 by shRNA severely reduced cell proliferation and survival [114].

Compared to controls, NIH3T3 fibroblasts stably expressing PIK3C3-H868R showed enhanced PIK3C3 lipid kinase activity and increased cell growth in vitro and increased tumor formation in mice [117]. The histopathological evaluation of tumor sections revealed the development of fibrosarcoma, supporting the role of PIK3C3 in regulating oncogenic transformation. The mechanism was mediated by the downregulation of TSC2 protein and the activation of RheB and mTOR/S6K1.

Although most studies revealed a positive correlation between PIK3C3 and cell proliferation and growth, one in vitro study of esophageal SCC showed the opposite results [92]. In a study using KYSE-150 and TE-12 esophageal SCC cells, PIK3C3 overexpression suppressed cancer cell proliferation [92]. Moreover, the addition of autophinib (a PIK3C3 inhibitor) significantly decreased apoptosis and increased the proliferation of oral SCC cells stably expressing the Per2 gene [109]. The above findings suggest a potential tumor suppressor role of PIK3C3 in oral and esophageal SCC. Further study is needed to clarify this.

### 5.2. Migration, Invasion, and Metastasis

Research on PIK3C3 in relation to migration, invasion, and metastasis is scarce (Table 1). Stable expression of PIK3C3 in MCF-7 breast cancer cells dramatically increased the intracellular accumulation of p62, the expression of epithelial–mesenchymal transition (EMT) markers (snail and vimentin), and the invasion of cells in vitro with activation of the MEK-ERK pathway [108]. In addition, inhibition of PIK3C3 reduced invasion and promoted the death of trametinib-resistant melanoma cells in an in vivo zebrafish xenograft of metastatic melanoma [112]. In contrast, PIK3C3 inhibits the invasion of HCC cells by regulating endosome-lysosome trafficking via the Rab7-RILP pathway [10]. Our study showed that inhibition of PIK3C3 by miR-338-5p suppresses autophagy and induces CRC cell EMT, migration, invasion, and metastasis [12] (Figure 3). The contradictory results may be due to the different cancer types and the experimentally manipulated mediators upstream of PIK3C3. More studies are needed to elucidate this issue.

## 6. Significance of PIK3C3 in Human Cancer

### 6.1. Cancer Stem Cells (CSCs)

Multiple studies have confirmed the essential role of PIK3C3/autophagy in the maintenance of CSCs. Autophagy-enriched HCT116 colon cancer cells induced by 5-FU treatment showed increased expression of CSC markers and 5-FU resistance [118]. Cotreatment with 36-077 (a PIK3C3 inhibitor) induced degradation of β-catenin protein with a reduction in the CSC population and significantly improved the efficacy of 5-FU treatment in HCT116 cells [118] (Figure 3). In a study using patient-derived leukemic stem cells, PIK-III (PIK3C3 inhibitor) inhibited autophagy induced by nilotinib (tyrosine kinase inhibitor) and reduced the number of leukemic stem cells in vitro and in vivo when used alone or in combination with nilotinib [119].

A marked increase in PIK3C3 protein was detected in HCC tissues and liver CSCs [106]. HCC cells stably expressing PIK3C3 revealed increased expression of stemness genes. RNA interference-mediated silencing of PIK3C3 had the opposite effect. Furthermore, Vps34-IN1 (a PIK3C3 inhibitor) suppressed the expression of stemness genes in HCC cells and inhibited the growth of tumors in vivo by activating AMP-activated kinase. However, the mechanism by which PIK3C3 regulates liver CSCs is independent of autophagy [106].

### 6.2. The Role of PIK3C3 in Cancer Immunity

The principal mechanism of cancer immunity is through function of CD8+ T cells, which kill cancer cells after recognizing specific tumor peptide-major histocompatibility complex class I complexes on target cells [120]. CD4+ T cells produce IFN-γ, contributing to an inflammatory environment that favors antitumor immunity [120]. Natural killer (NK) cells also play a major role in that they directly kill cancer cells using cytotoxic molecules.

Autophagy activation in tumor cells might play a major role in impairing the an-titumor immune response. Targeting autophagy-related genes inhibits tumor growth and improves T cell and NK-mediated killing [121,122]. In CT26 colon cancer cells and B16-F10 melanoma cells, targeting PIK3C3 with siRNA or PIK3C3 inhibitors reduced autophagy and tumor growth and improved mouse survival by inducing the infiltration of natural killer, CD8+, and CD4+ T effector cells [105]. Such infiltration resulted from the induction of STAT1 and IRF7, and the expression and secretion of pro-inflammatory chemokines and cytokines (CCL5, CXCL10, and interferon-γ) in tumor cells in vitro and tumor plasma in vivo [105].

### 6.3. Expression of PIK3C3 in Human Cancer

The expression of PIK3C3 RNA in human cancer in The Cancer Genome Atlas (TCGA) database was analyzed by UALCAN (http://uacan.path.uab.edu/ (accessed on 8 July 2021)) [123] and OncoLnc (http://www.oncolnc.org/ (accessed on 8 July 2021)) [124]. The results are summarized in Table 2. Significantly upregulated PIK3C3 RNA expression was observed in HCC, cholangiocarcinoma, gastric adenocarcinoma, and SCC of the lung, head, and neck versus the corresponding normal controls. On the other hand, PIK3C3 RNA was significantly downregulated in colon and rectum adenocarcinoma, breast invasive carcinoma, renal cell carcinoma (RCC), prostate adenocarcinoma, thyroid carcinoma, and endometrial carcinoma (Table 2).

PIK3C3 RNA levels were increased in high versus low tumor stages of HCC, gastric adenocarcinoma, and uterine carcinosarcoma. In contrast, PIK3C3 levels were decreased in high tumor stages of renal clear cell carcinoma, endometrial carcinoma, thyroid carcinoma, and mesothelioma (Table 2). PIK3C3 RNA levels were higher in grade 2–4 head and neck SCC samples but lower in grade 3 and 4 renal clear cell carcinoma samples than in grade 1 samples for each cancer. In terms of histologic type, the PIK3C3 RNA levels were higher in colon mucinous carcinoma but lower in cervical mucinous carcinoma than in conventional-type adenocarcinoma for each cancer (Table 2).

PIK3C3 protein levels were significantly elevated in highly tumorigenic breast cancer cell lines (MDA-MB-231, MDA-MB-468, SKBR-3) compared with those with relatively low tumorigenic potential (MCF-7 and T47D) [125]. Published research on PIK3C3 expression in human cancer tissues is scarce. Both increased and decreased levels of PIK3C3 protein have been reported in HCC [10,106]. In a series of 31 pairs of normal and esophageal SCC tissues, PIK3C3 mRNA levels were significantly lower in esophageal SCC tissues [92].

### 6.4. Prognostic Significance of PIK3C3 in Human Cancer

In the TCGA analysis of the UALCAN database (http://uacan.path.uab.edu/ (accessed on 8 July 2021)) [123], high expression of PIK3C3 RNA was correlated with poor patient survival for breast carcinoma, gastric carcinoma, HCC, mesothelioma, and low-grade glioma but associated with improved survival for renal clear cell carcinoma (Table 2).

According to published research, patients with high PIK3C3 protein expression in HCC tumors displayed worse overall survival. PIK3C3 protein levels in HCC tissues were positively correlated with tumor stage [106]. A study of 149 chronic lymphocytic leukemia cases revealed that high PIK3C3 expression assessed by real-time RT-PCR was independently associated with poor survival [126]. In contrast, esophageal SCC patients with low PIK3C3 protein levels exhibited a worse prognosis than those patients with high PIK3C3 levels [92].

## 7. PIK3C3 as Therapeutic Target

Autophagy maintains cell survival when modifications occur in the cellular environment and helps tumor cells resist metabolic stress and cancer treatment [127]. PIK3C3 is one of the druggable targets in the signaling pathways that regulate the autophagy machinery. Inhibition of PIK3C3 successfully inhibits autophagy, making the complex an attractive target for developing autophagy inhibitors [127]. In addition, PIK3C3 could induce oncogenic transformation and enhance tumor cell proliferation and invasion through mechanisms independent of autophagy [106,108,111,117]. Novel selective PIK3C3 inhibitors have been developed in recent years. The details are summarized in Table 3 and described as follows.

### 7.1. SAR405

SAR405, identified by the Pasquier group, is a highly potent and selective inhibitor of PIK3C3 [129]. SAR405 is a compound of the (2S)-tetrahydropyrimido-pyrimidinones series and has a half-maximal inhibitory concentration (IC50) of 1 nM in the phosphorylation of a PtdIns substrate by the human recombinant PIK3C3 enzyme [129]. SAR405 affects vesicle trafficking between late endosomes to lysosomes, resulting in an impairment of lysosomal function. SAR405 is able to inhibit autophagy induced by starvation or by the catalytic mTOR inhibitor AZD8055 [129]. SAR405 has been reported to inhibit the tumor growth or proliferation of CRC [105], melanoma [105], breast cancer [107], and head and neck SCC [110] when used alone and to synergize with cisplatin [110,128], lapatinib [107], BYL719 [107], everolimus [129], and anti-PD-L1/PD-1 immunotherapy [105].

In two HER2+ cell lines, BT474 and SKBR3, SAR405 significantly increased caspase-3/7 activity (apoptosis) and enhanced the growth inhibition mediated by lapatinib (dual inhibitors of HER2 and EGFR) and BYL719 (a type I PI3K inhibitor) [107]. In models of CT26 CRC cells and B16-F10 melanoma cells, targeting PIK3C3 with SAR405 inhibited autophagy, significantly decreased tumor growth, and improved mouse survival by inducing the infiltration of natural killer, CD8+, and CD4+ T effector cells [105]. Such infiltration switched cold immune-desert tumors into hot, inflamed, immune-infiltrated tumors. In addition, adding SAR405 enhanced the efficacy of anti-PD-L1/PD-1 immunotherapy and prolonged the survival of tumor-bearing mice [105].

### 7.2. PIK-III

PIK-III, identified by the Murphy group, is a selective PIK3C3 inhibitor of the bisaminopyrimidine family [131]. Biochemical profiling showed that PIK-III is at least 100-fold selective for VPS34 over related lipid kinases such as PI3Kα and mTOR [131]. PIK-III inhibits autophagy, the clearance of mitochondria, and the degradation of autophagy substrate receptors, such as p62, NBR1, and NDP52 [131]. PIK-III has been reported to inhibit tumor growth or invasion in CRC [103] and melanoma [112] when used alone and to synergize with sunitinib, erlotinib [13], nilotinib [119], and trametinib [112].

In an RKO CRC cell model, inhibition of PIK3C3 using PIK-III enhanced lysosomal degradation of transferrin, leading to impaired iron uptake and cell growth, which was partially rescued by excess iron [103]. In an A375 melanoma cell model, PIK-III used alone or combined with trametinib (MEK inhibitor) reduced the invasive and metastatic potential of tumor cells in an in vivo zebrafish xenograft model [112]. In addition, PIK-III reduced the numbers of chronic myeloid leukemia stem cells, suppressed autophagy induced by nilotinib (a tyrosine kinase inhibitor), and sensitized these cells to nilotinib in vitro and in vivo [119]. In two breast cancer cell lines, MDA-MB-231 and MCF-7, PIK-III significantly potentiated the cytotoxicity of sunitinib and erlotinib [13].

### 7.3. VPS34-IN1

The bis-aminopyrimidine compound VPS34-IN1, identified by the Alessi group, is a highly potent and selective inhibitor of PIK3C3 [132]. VPS34-IN1 inhibits PIK3C3 with a 25 nM IC50 in vitro but does not significantly inhibit the activity of class I and class II PI3Ks [132]. VPS34-IN1 has been reported to inhibit tumor growth or cell survival in HCC [106], glioblastoma [113], and myeloid leukemia [130] when used alone and to synergize with ZSTK474 [106], vemurafenib [113], L-asparaginase [130], and ceritinib [11].

PIK3C3 upregulation facilitates liver CSC expansion in HCC. PIK3C3 inhibition by VPS34-IN1 effectively eliminates liver CSCs and inhibits the growth of tumors in vivo [106]. VPS34-IN1 abrogates the expansion of liver CSCs induced by ZSTK474 (a PI3K inhibitor). The combination of VPS34-IN1 and ZSTK474 results in a more potent inhibitory effect on stemness genes in vitro and in vivo than VPS34-IN1 alone [106].

In a study using AM38 glioblastoma cells and patient-derived MAF794 ganglioglioma cells, PIK3C3 inhibition by VPS34-IN1 reduced autophagic flux and decreased cell survival in a dose-dependent manner [113]. The addition of VPS-IN1 significantly improves the response to vemurafenib (a BRAF inhibitor) [113].

VPS34-IN1 was found to induce apoptosis in nine acute myeloid leukemia (AML) cell lines but not in normal CD34+ hematopoietic cells [130]. VPS34-IN1 inhibits basal and L-asparaginase-induced autophagy in AML cells and is synergistic with L-asparaginase [130]. In MOLM-14 leukemic cells, VPS34-IN1 was also found to impair vesicular trafficking and mTORC1 signaling and inhibit STAT5 phosphorylation downstream of FLT3-ITD signaling [130]. The mechanism controlling FLT3-ITD signaling by PIK3C3 provides important insight into the oncogenesis of AML and could be a new therapeutic strategy.

### 7.4. Other Selective PIK3C3 Inhibitors

SB02024, developed by the Tamm group, is a potent selective inhibitor of PIK3C3. SB02024 was found to suppress autophagy and tumor growth in CRC, melanoma [105], and breast cancer [13] and to improve the therapeutic benefit of anti-PD-L1/PD-1 immunotherapy [105], sunitinib, and erlotinib [13]. Singh et al. synthesized 36-077, a specific and potent inhibitor of PIK3C3/VPS34 kinase. Co-treatment with 36-077 improves the efficacy of 5-FU, and suppressed GSK-3β/Wnt/β-catenin signaling to inhibit CSC population in HCT116 cells [118].

## 8. Conclusions and Future Directions

PIK3C3 is a critical regulator of autophagy and vesicle trafficking which could maintain cell survival when modifications occur in the cellular environment and could help tumor cells resist metabolic stress and cancer treatment [127]. In addition, PIK3C3 could induce oncogenic transformation and enhance tumor cell proliferation, growth, and invasion through mechanisms independent of autophagy [106,108,111,117], while its role in cancer invasion and metastasis needs to be elucidated. In some cancer types, PIK3C3 may play a tumor suppressor role, such as esophageal and oral SCCs [92,109]. Novel selective PIK3C3 inhibitors, such as SAR405, PIK-III, and VPS34-IN1, have been developed in recent years. Most studies have shown that PIK3C3 inhibitors can repress tumor growth and synergize with various anticancer drugs. Altogether, PIK3C3 is a potential therapeutic target for cancer, and targeting agents should be designed with the tumor microenvironment and specific cancer type in mind. More studies are needed to clarify the clinical and prognostic significance of PIK3C3 in different cancer types.

## Figures and Tables

**Figure 1 ijms-22-10964-f001:**
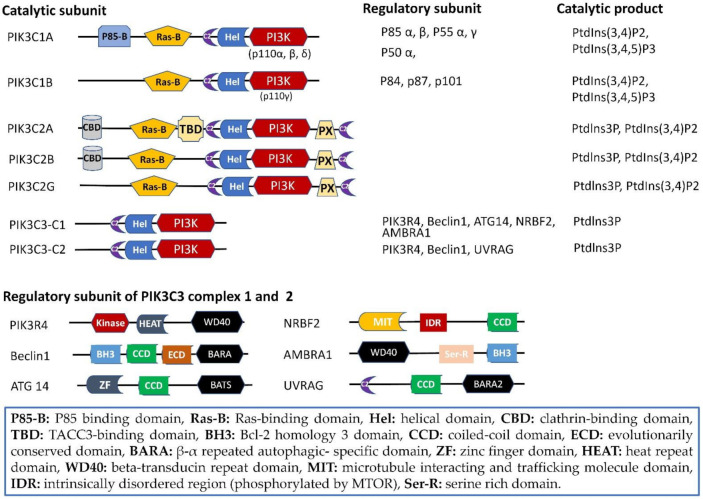
Structure of the catalytic and regulatory subunits and catalytic products of PI3K family proteins.

**Figure 2 ijms-22-10964-f002:**
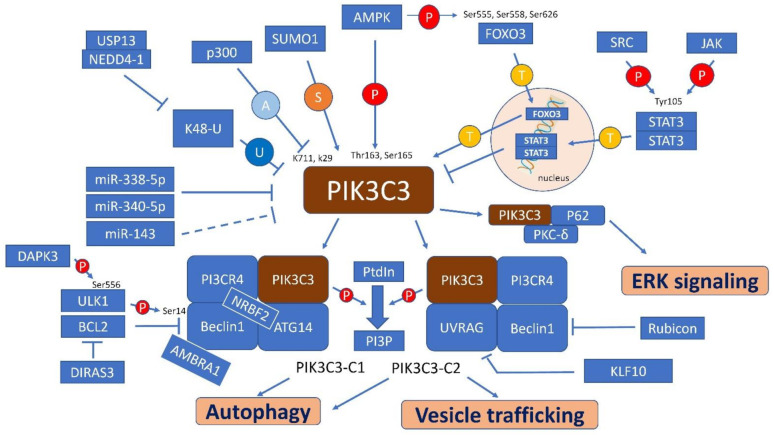
PIK3C3 related pathway in mammals. P: phosphorylation; A: acetylation; T: nuclear entry and regulation by transcription; U: ubiquitination; S: SUMOylation.

**Figure 3 ijms-22-10964-f003:**
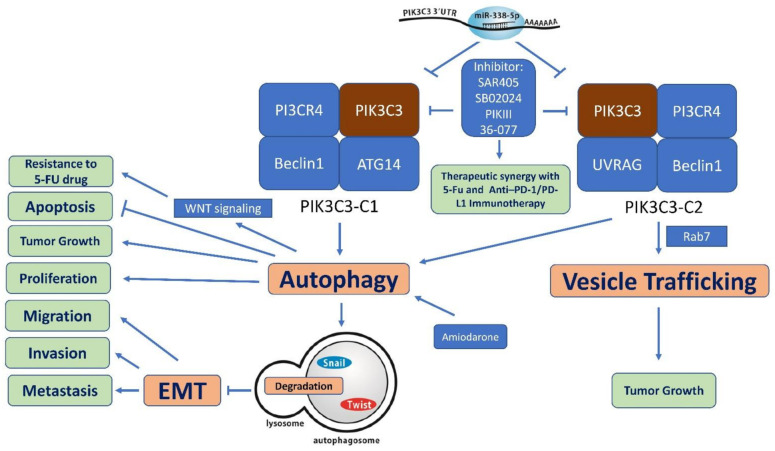
Biological role of PIK3C3 in the tumorigenesis of colorectal cancer. PIK3C3 complex 1 (PIK3C3-C1) is required to initiate autophagy, which induces tumor growth, proliferation, and resistance to 5-FU. PIK3C3 complex 2 (PIK3C3-C2) promotes tumor growth by regulating Rab7. Formation of the PIK3C3 complexes can be inhibited by miR-338-5p and PIK3C3 inhibitors. MiR-338-5p inhibits PIK3C3, and then suppresses the degradation of Snail and Twist by autophagy, resulting in tumor cell migration, invasion, and metastasis [12]. PIK3C3 inhibitors inhibit tumor proliferation and growth, and exhibit therapeutic synergy with 5-FU and anti-PD-1/PD-L1 immunotherapy.

**Table 1 ijms-22-10964-t001:** The biological function of PIK3C3 in cancer.

Cancer Type	Cell Line	Role of PIK3C3	Proliferation	Growth	Migration	Invasion	Metastasis	Apoptosis	Genetic Manipulation	PIK3C3 Inhibitor	Reference
CRC	RKO	Onco	Indu	Indu	-	-	-	-	KD	PIK-III	[103]
CRC	HCT116	Onco	-	-	-	-	-	Inhi	KD	-	[104]
CRC	CT26	Onco	-	Indu	-	-	-	-	KD	SB02024 SAR405	[105]
CRC	SW480	Sup	-	-	Inhi	Inhi	Inhi	-	Expression	-	[12]
HCC	Huh7, MHCC97H	Onco	-	Indu	-	-	-	-	KD	Vps34-IN1	[106]
HCC	HepG2, SMMC-7721	Sup	-	-	-	Inhi	-	-	Expression, KD	-	[10]
Breast carcinoma	BT474, SKBR3	Onco	Indu	-	-	-	-	-	KD	SAR405	[107]
Breast carcinoma	MCF-7, MDA-MB-231	Onco	-	Indu	-	-	-	-	-	SB02024	[13]
Breast carcinoma	MCF-7	Onco	-	Indu	-	Indu	-	-	Expression, KD	-	[108]
Esophageal SCC	KYSE-150, TE-12	Sup	Inhi	-	-	-	-	-	Expression	-	[92]
Oral SCC	TSCCA	Sup	Inhi	-	-	-	-	Indu	-	Autophinib	[109]
Head and neck SCC	UM-SCC-1	Onco	-	Indu	-	-	-	-	-	SAR405	[110]
Lung adenocarcinoma	H1975	Onco	-	-	-	-	-	Inhi	KD	-	[111]
Melanoma	B16-F10	Onco	-	Indu	-	-	-	-	KD	SB02024 SAR405	[105]
Melanoma	A375	Onco	-	-	-	Indu	-	-	-	PIK-III	[112]
Glioblastoma	AM38	Onco	-	Indu	-	-	-	-	-	VPS34-IN1	[113]
Acute lymphocytic leukemia	REH, REHS1	Onco	Indu	-	-	-	-	-	KD	-	[114]

CRC: colorectal cancer; Indu: Induce; Inhi: Inhibit; KD: knockdown; Onco: oncogenic; SCC: squamous cell carcinoma; Sup: tumor suppresser; - no data.

**Table 2 ijms-22-10964-t002:** Expression pattern of PIK3C3 in human cancer according to the TCGA database.

	UALCAN						OncoLnc ^#^
Cancer Type	Expression (vs. Normal)	Tumor Stage (vs. Stage 1)	N Stage (vs. N0)	Tumor Grade (vs. Grade 1)	Mucinous Type (vs. Conventional Type)	Poor Overall Survival	Poor Overall Survival
Colon adenocarcinoma	Down ***	ND	ND	-	Up *	ND	ND
Rectal adenocarcinoma	Down ***	ND	ND	-	ND	ND	ND
Adrenocortical carcinoma	-	ND	ND	-	-	ND	-
Acute myeloid leukemia	-	--	-	-	-	ND	ND
Bladder urothelial carcinoma	ND	ND	ND	-	-	ND	Up *
Low-grade glioma	-	-	-	ND	-	Up **	Up *
Breast invasive carcinoma	Down ***	ND	ND	-	ND	Up *	ND
Cervical carcinoma	ND	ND	ND	ND	Down * (mucinous vs. SCC)	ND	Up *
Cholangiocarcinoma	Up ***	ND	ND	ND	-	ND	-
Esophageal carcinoma	ND	ND	ND	ND	-	ND	ND
Glioblastoma	ND	-	-	--	-	ND	ND
Head and neck SCC	Up ***	ND	ND	Up (grade 2 *, 3 * & 4 ***)	-	ND	ND
Renal clear cell carcinoma	Down ***	Down *** (stage 3, 4)	ND	Down (grade 3 * & 4 **)	-	Down ***	Down **
Renal papillary cell carcinoma	Down ***	ND	ND	-	-	ND	ND
HCC	Up ***	Up ** (stage 3)	Up *	ND	-	Up **	ND
Lung adenocarcinoma	ND	ND	ND	-	ND	ND	ND
Lung SCC	Up ***	ND	ND	-	-	ND	ND
Diffuse large B cell lymphoma	-	ND	-	-	-	ND	-
Mesothelioma	-	Down * (stage 2)	ND	-	-	Up *	-
Ovarian serous carcinoma	-	--	-	ND	-	ND	ND
Pancreatic adenocarcinoma	ND	ND	ND	ND	-	ND	ND
Prostate adenocarcinoma	Down ***	-	ND	ND	-	ND	-
Skin melanoma	ND	ND	ND	-	-	ND	Down *
Stomach adenocarcinoma	Up ***	Up * (stage 4)	ND	ND	ND	Up *	ND
Thyroid carcinoma	Down ***	Down ** (stage 4)	ND	-	-	ND	-
Endometrial carcinoma	Down **	Down * (stage 3)	-	-	-	ND	ND
Uterine carcinosarcoma	-	Up * (stage 3)	-	-	-	ND	

ND: no statistical difference; SCC: squamous cell carcinoma; HCC: hepatocellular carcinoma; * *p* < 0.05; ** *p* < 0.01; *** *p* < 0.001; - no data. # In the analysis, up-regulated PIK3C3 was defined as “above the median of PIK3C3 expression”.

**Table 3 ijms-22-10964-t003:** Summary of the selective PIK3C3 inhibitors and their effects on cancer.

PIK3C3 Inhibitor	Cancer	Cell Line	Effect (When Used Alone)	Synergistic Drug	Reference
SAR405	CRC	CT-26	Inhibits tumor growth	Anti-PD-1/PD-L1 immunotherapy	[105]
	Melanoma	B16-F10	Inhibits tumor growth	Anti-PD-1/PD-L1 immunotherapy	[105]
	Breast carcinoma	BT474, SKBR3	Inhibit proliferation	Lapatinib, BYL719	[107]
	Head & neck SCC	UM-SCC-1	Inhibits tumor growth	Cisplatin	[110]
	Bladder carcinoma	RT-112	-	Cisplatin	[128]
	Lung large cell carcinoma	H1299	-	Everolimus	[129]
	Renal cell carcinoma	ACHN, 860-O	-	Everolimus	[129]
PIK-III	CRC	RKO	Inhibits proliferation and tumor growth	-	[103]
	Acute myeloid leukemia	MOLM14, MV411, OCI-AML2, HL60, THP1, K562	Induces cell death	-	[130]
	Melanoma	A375	Inhibits invasion	Trametinib	[112]
	Breast carcinoma	MCF-7, MDA-MB-231	-	Sunitinib, erlotinib	[13]
	Chronic myeloid leukemia	Leukemic cells from mice	-	Nilotinib	[119]
VPS34-IN1	HCC	Huh7, MHCC97H	Inhibits tumor growth	ZSTK474	[106]
	Glioblastoma	AM38	Inhibits tumor growth	Vemurafenib	[113]
	Acute myeloid leukemia	MOLM14	Induces cell death	L-asparaginase	[130]
	Lung adenocarcinoma	H3122	-	Ceritinib	[11]
SB02024	CRC	CT-26	Inhibits tumor growth	Anti-PD-1/PD-L1 immunotherapy	[105]
	Melanoma	B16-F10	Inhibits tumor growth	Anti-PD-1/PD-L1 immunotherapy	[105]
	Breast carcinoma	MCF-7, MDA-MB-231	Inhibits tumor growth	Sunitinib, erlotinib	[13]
Autophinib	Acute myeloid leukemia	MOLM14, MV411, OCI-AML2, HL60, THP1, K562	Induces cell death	-	[130]
	Oral SCC	TSCCA	Increases proliferation and decreases apoptosis	-	[109]
36-077	CRC	HCT116	-	5-FU	[118]

CRC: colorectal cancer; HCC: hepatocellular carcinoma; SCC: squamous cell carcinoma; - no data.

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
