# Peer review of "The Role of Phosphatidylinositol 3-Kinase Catalytic Subunit Type 3 in the Pathogenesis of Human Cancer"

_ijms, 2021, doi:10.3390/ijms222010964_

Round 1

Reviewer 1 Report

Major comments:

  1. If autophagy contributes both suppression and promotion of carcinogenesis (Line 48-49), how to sure PIK3C3-mediated autophagy majorly mediates cell survival (Line 29).
  2. I suggest authors delete the discussion in non-tumor parts since the title reveals the focus on tumor pathogenesis. It is too hard to follow and cannot compare together.
  3. In Line 104, how can a gene such as PIK3C2A have both opposite functions? Please describe more clearer and the detailed mechanism should be shown.
  4. In Line 107, PIK3C2A and PIK3C2B both promote tumor progression, but how to modulate cancer death and why IL6 should be described here?
  5. Please indicate the mechanism why the three types of PIK3C2s has different roles in tumors in Line 115. Is that because of the different domains presented such as CBD and TBD. And what is the role of PIK3C2 autophagy determining the different functions?
  6. I cannot fully understand the description of BECN1 in Line 141 to 162. How can BECN1 involves in cancer progression (Line 141) but inhibits CRC migration and metastasis (Line 152)? How can activation of PIK3C3C1 by DAPK3 suppresses tumor progression (Line 155) but activation by PTP4A3 promote tumor progression (Line 157)? Please clearly reveal the mechanism.
  7. For example, in Line 132 please clearly use such as “increase” or “decrease” to replace “regulate”. Readers cannot know it is positive or negative regulation. Please check such kind of problem in the article.
  8. Please reveal more detail for ATG14 and AMPRA1 in Line 164 to 168.
  9. Why do you suddenly indicate c-VPS and Rab7A in Line 184. They are not shown in Figure 2?
  10. Please indicate why and how UVRAG has the opposite role in tumor progression (Line 185 to 195). If it induces tumor progression through CDK and integrin signaling, why it plays tumor suppressor in melanoma? And the mechanism for UVRAG in radioresistance and radiosensitivity. The description should be more logical and detailed.
  11. Please indicate the mechanism mediating PIK3C3 expression in section 3 (Line 196). The transcriptional factor, epigenetic regulation, and so on. The current description is not necessary.
  12. How does STAT3 suppress PIK3C3 expression since STAT3 is a transcriptional factor (Line 226).
  13. How to induce immune cell infiltration by inhibition of PIK3C3 (Line 267)?
  14. Please describe more about how PIK3C3 inhibits ARF in Line 270?
  15. Please indicate the detailed condition in Line 284 describing the opposite role of PIK3C3 in a tumor. Reference 106 is PIK3CA (Line 286), but this article is PIK3C3, is it the same gene?
  16. Please propose your opinion in section 5.2 on why inhibition of PIK3C3 induces migration, invasion, and metastasis if PIK3C3 promotes tumor progression.
  17. The content of section 6.1 is not carcinogenesis of PIK3C3. It is expression levels. Please reorganize with section 6.3.
  18. Please reveal the detailed mechanism of how PIK3C3 increases GSK-3β-Wnt-β-catenin (section 6.2).
  19. If PIK3C3 inhibitor Vps34-IN1 suppresses cancer stemness via activating AMPK (Line 342), does it mean that PIK3C3 suppresses AMPK in the tumor? But there is AMPK also activates PIK3C3 in Figure 2. How to explain the phenomena?
  20. I suggest you delete section 7. It is too complicated if you compare it with other diseases since the pathogenesis is so different.

Miner comments:

  1. Line 29 in Abstract, “Inhibition of PIK3C3 could successfully inhibit autophagy, which…” here which means autophagy or inhibition of PIK3C3. Some question also appears in Line 53, Line 59.
  2. PIK3R4 In Line 54, 126, 127 means PIK3CR4 in Figure 1?
  3. In Line 60, UVRAG means UV radiation resistance-associated protein? If yes, you miss ‘protein”.
  4. Does AMBRA1 in Line 132 mean AMBR1 in Figure 2?
  5. Please indicate the specific tumor types in Line 515 whereas PIK3C3 as a tumor suppressor.

Author Response

Dear Sir/Madam:

We are grateful for the opportunity to improve the quality of our article, and for the helpful comments. Our manuscript was revised according to these comments. The revised parts were highlighted in red. We deleted section 7 (non-neoplastic diseases), and merged the information of section 6.1 (carcinogenesis) in section 5.1 and 6.3. We added a new section 6.2 (the role of PIK3C3 in cancer immunity). The following are our point-by-point responses to the comments.

Major comments:

  1. If autophagy contributes both suppression and promotion of carcinogenesis (Line 48-49), how to sure PIK3C3-mediated autophagy majorly mediates cell survival (Line 29).

Response: As shown in Table 1, the role of PIK3C3 might be different in different cell types. Therefore, it is important to choose the appropriate cancer type for anti-PIK3C3 therapy.

  1. I suggest authors delete the discussion in non-tumor parts since the title reveals the focus on tumor pathogenesis. It is too hard to follow and cannot compare together.

Response: Thank you for the suggestion. We deleted the section.

  1. In Line 104, how can a gene such as PIK3C2A have both opposite functions? Please describe more clearer and the detailed mechanism should be shown.

Response: Reduction of PIK3C2A in breast cancer models initially impairs tumor growth but later leads to the convergent evolution of fast-growing clones with mitotic checkpoint defects. We added the description in the 2nd paragraph of section 2.2.

  1. In Line 107, PIK3C2A and PIK3C2B both promote tumor progression, but how to modulate cancer death and why IL6 should be described here?

Response: In human lung carcinoma cells, increased expression of PIK3C2A and B are associated with talc particle-induced cell death and IL-6 secretion. These two effects are key cellular events leading to fibrosis. We added the description in the 2nd paragraph of section 2.2. 

  1. Please indicate the mechanism why the three types of PIK3C2s has different roles in tumors in Line 115. Is that because of the different domains presented such as CBD and TBD. And what is the role of PIK3C2 autophagy determining the different functions?

Response: Only PIK3C2A regulates autophagosome biogenesis by PI(3)P-mediated pathway. On the other hand, The N-terminal extension of PIK3C2A, which has specific roles in the activation of enzymes, possesses a clathrin-binding region and TACC3-binding region. PIK3CA can mediate Clathrin and TACC, and regulate cell mitosis. In a different way, the N-terminal extension of PIK3C2A and PIK3C2B possesses clathrin binding properties. PIK3C2A and B can regulate endosomal trafficking and cell migration through mediating clathrin. Altogether, the three types of PIK3C2 play different roles in human diseases, especially in epithelial carcinogenesis. It may be due to the different domains. We added the description in section 2.2.

  1. I cannot fully understand the description of BECN1 in Line 141 to 162. How can BECN1 involves in cancer progression (Line 141) but inhibits CRC migration and metastasis (Line 152)? How can activation of PIK3C3C1 by DAPK3 suppresses tumor progression (Line 155) but activation by PTP4A3 promote tumor progression (Line 157)? Please clearly reveal the mechanism.

Response:

BECN1 involves in cancer progression by the following mechanisms: (a) Enhanced autophagosome formation mediated by PTP4A3, an activator of PIK3C3-BECN1 complex, promotes the proliferation of ovarian cancer cells. (b) Reduced BECN1/Vps15/Vps34 complex activity mediated by the interaction of BECN1 with androgen receptor can inhibit ERK-mediated growth factor signaling and suppress the growth of castration-resistant prostate cancer.

On the other hand, RNF216 induces colorectal cancer cell proliferation and migration by inhibiting BECN1-mediated autophagy, suggesting a tumor suppressor role in colorectal cancer. The role of BECN1 may be different in different cancer types.

The results about PIK3C3C1 regulated by DAPK3 and PTP4A3 were from two separate studies. We described the mechanisms in the 2nd paragraph of 2.3.1.

  1. For example, in Line 132 please clearly use such as “increase” or “decrease” to replace “regulate”. Readers cannot know it is positive or negative regulation. Please check such kind of problem in the article.

Response: Thank you for the suggestion. We revised the text.

  1. Please reveal more detail for ATG14 and AMPRA1 in Line 164 to 168.

Response: We added more details in the 1st paragraph of section 2.3.1.

  1. Why do you suddenly indicate c-VPS and Rab7A in Line 184. They are not shown in Figure 2 ?

Response: We apologize for this mistake. We deleted the reference of figure 2 in the sentence.

  1. Please indicate why and how UVRAG has the opposite role in tumor progression (Line 185 to 195). If it induces tumor progression through CDK and integrin signaling, why it plays tumor suppressor in melanoma? And the mechanism for UVRAG in radioresistance and radiosensitivity. The description should be more logical and detailed.

Response: Thank you for the comment. We reorganized this part, and deleted some inappropriate references. Please see the 2nd and 3rd paragraphs of section 2.3.2.

  1. Please indicate the mechanism mediating PIK3C3 expression in section 3 (Line 196). The transcriptional factor, epigenetic regulation, and so on. The current description is not necessary.

Response: We apologize for the misleading heading of section 3. The heading was revised as “Expression of PIK3C3 in human tissues”. The mechanisms mediating PIK3C3 expression were described in section 4, including acetylation, transcription, posttranslational ubiquitination, modification, and miRNA regulation.

  1. How does STAT3 suppress PIK3C3 expression since STAT3 is a transcriptional factor (Line 226).

Response: We apologize for this. It is not clear whether STAT3 is a transcriptional factor of PIK3C3. We revised the text in the first paragraph of section 4.

  1. How to induce immune cell infiltration by inhibition of PIK3C3 (Line 267)?

Response: According to the suggestion of another reviewer, we added a new section 6.2 (the role of PIK3C3 in cancer immunity). The mechanism of immune cell infiltration by PIK3C3 inhibition was described in the new section 6.2.

  1. Please describe more about how PIK3C3 inhibits ARF in Line 270?

Response: We added more information in the 3rd paragraph of section 5.1.

  1. Please indicate the detailed condition in Line 284 describing the opposite role of PIK3C3 in a tumor. Reference 106 is PIK3CA (Line 286), but this article is PIK3C3, is it the same gene?

Response: We described the opposite role of PIK3C3 in esophageal SCC cells in more details in the 7th paragraph of section 5.1.

No, PIK3CA and PIK3C3 are not the same gene. As we described in the text, autophinib (a PIK3C3 inhibitor) significantly decreased apoptosis and increased the proliferation of oral SCC cells stably expressing the Per2 gene. It suggests that a potential tumor suppressor role of PIK3C3 in oral SCC.

  1. Please propose your opinion in section 5.2 on why inhibition of PIK3C3 induces migration, invasion, and metastasis if PIK3C3 promotes tumor progression.

Response: The contradictory results are from different studies using different cell lines (Table 1). The role of PIK3C3 might be different in different cell types. The other reason is that PIK3C3 is central to autophagy. Autophagy is reported to have dual, contradictory roles in carcinogenesis, and is stage dependent. Autophagy may reduce tumor progression in early stage of tumor, but help tumor cells resist metabolic stress and cancer treatment in late stage.

  1. The content of section 6.1 is not carcinogenesis of PIK3C3. It is expression levels. Please reorganize with section 6.3.

Response: Thank you for the suggestion. We deleted section 6.1 carcinogenesis, and merge the information in sections 5.1 and 6.3.

  1. Please reveal the detailed mechanism of how PIK3C3 increases GSK-3β-Wnt-β-catenin (section 6.2).

Response: Thank you for the comment. The reference article (Ref. 117. PIK3C3 inhibition promotes sensitivity to colon cancer therapy by inhibiting cancer stem cells. Cancers (Basel) 2021, 13, 2168.) showed that cotreatment of 5-FU and 36-077 (a PIK3C3 inhibitor) inhibited expression of p-GSK-3β (Ser9), and induced degradation of β-catenin protein without explanation or discussion of the relationship between the molecules. The decreased GSK-3β seems not to explain the degradation of β-catenin. Therefore, we deleted the part of GSK-3β in the first paragraph of section 6.1.

  1. If PIK3C3 inhibitor Vps34-IN1 suppresses cancer stemness via activating AMPK (Line 342), does it mean that PIK3C3 suppresses AMPK in the tumor ? But there is AMPK also activates PIK3C3 in Figure 2. How to explain the phenomena ?

Response: There is evidence that genetic or pharmacologic inhibition of PIK3C3 leads to AMPK activation. Thus it is possible that PIK3C3 may suppress AMPK in the tumor. Further studies are needed to confirm this. The phenomenon that AMPK activates PIK3C3 is like a negative feedback.

  1. I suggest you delete section 7. It is too complicated if you compare it with other diseases since the pathogenesis is so different.

Response: Thank you for the suggestion. We deleted the section.

Miner comments:

  1. Line 29 in Abstract, “Inhibition of PIK3C3 could successfully inhibit autophagy, which…” here which means autophagy or inhibition of PIK3C3. Some question also appears in Line 53, Line 59.

Response: Thank you for the comment. We revised the text.

  1. PIK3R4 In Line 54, 126, 127 means PIK3CR4 in Figure 1?

Response: Yes, they are the same. We apologize for this, and remade the figure.

  1. In Line 60, UVRAG means UV radiation resistance-associated protein? If yes, you miss ‘protein”.

Response: Thank you for the comment. We revised the text.

  1. Does AMBRA1 in Line 132 mean AMBR1 in Figure 2?

Response: Yes, we apologize for this, and remade the figure.

  1. Please indicate the specific tumor types in Line 515 whereas PIK3C3 as a tumor suppressor.

Response: Thank you for the comment. We added it in section 8.

Reviewer 2 Report

Dear authors,

The manuscript entitled "“The Role of Phosphatidylinositol 3-Kinase Catalytic Subunit 2 Type 3 in the Pathogenesis of Human Cancer”" could be interesting for the scientific and clinical community.

In order to improve the quality of the article, the authors are recommended to make the changes below:

  1. The authors are recommended to improve the writing of the article. It should be more fluid.
  2. In Figure 1, all the names associated with the acronyms of the domains, used in the figure, should be show at the foot of the figure.
  3. In line 88, bibliographic citations [8,21-27], should be deleted.
  4. In section 2.3.1, line 166, “carcinomas of the pancreas, ovary, colorectum and liver ”, should be modified to “pancreatic, ovarian, colorectal and liver cancer”.
  5. In Table 1, long headings should be fit into the columns using auto-fit or angling the text.
  6. The authors are recommended to reference Table 1 in the text.
  7. The biological functions described for PIK3C3 in the article, are closely related to carcinogenesis. Because of this, sections 5 and 6 must be merged.
  8. The role of PIK3C3 in cancer immunity should be described in section 6.
  9. Section 7 is not related to the title and subject of the article. Authors are advised to delete section 7.

Kind regards.

Author Response

Dear Sir/Madam:

We are grateful for the opportunity to improve the quality of our article, and for the helpful comments. Our manuscript was revised according to these comments. The revised parts were highlighted in red. We deleted section 7 (non-neoplastic diseases), and merged the information of section 6.1 (carcinogenesis) in section 5.1 and 6.3. We added a new section 6.2 (the role of PIK3C3 in cancer immunity). The following are our point-by-point responses to the comments.

  1. The authors are recommended to improve the writing of the article. It should be more fluid.

Response: Thank you for the suggestion. We will keep improving our writing. The manuscript was edited by Nature Research Editing Service.

  1. In Figure 1, all the names associated with the acronyms of the domains, used in the figure, should be show at the foot of the figure.

Response: Thank you. We showed them at the foot of figure 1.

  1. In line 88, bibliographic citations [8,21-27], should be deleted.

Response: Thank you. We deleted the citations.

  1. In section 2.3.1, line 166, “carcinomas of the pancreas, ovary, colorectum and liver ”, should be modified to “pancreatic, ovarian, colorectal and liver cancer”.

Response: Thank you. We revised the text.

  1. In Table 1, long headings should be fit into the columns using auto-fit or angling the text.

Response: Thank you for the suggestion. We will communicate with the editors to adjust the table.

  1. The authors are recommended to reference Table 1 in the text.

Response: We had referenced Table 1 in the text.

  1. The biological functions described for PIK3C3 in the article, are closely related to carcinogenesis. Because of this, sections 5 and 6 must be merged.

Response: Thank you for the suggestion. We deleted section 6.1 (carcinogenesis), and merged the information in section 5.1 and 6.3. 

  1. The role of PIK3C3 in cancer immunity should be described in section 6.

Response: Thank you for the suggestion. We described it in the new section 6.2.

  1. Section 7 is not related to the title and subject of the article. Authors are advised to delete section 7.

Response: Thank you for the suggestion. We deleted the section.

Round 2

Reviewer 1 Report

I have no more questions for this study

Author Response

Thank you for the reviewing and the helpful comments.

Reviewer 2 Report

Dears authors,

Thanks for following the recommendations. 

I would like to suggest just one modification to the last version of the article  entitled "The Role of Phosphatidylinositol 3-Kinase Catalytic Subunit Type 3 in the Pathogenesis of Human Cancer", before being published in the International Journal of Molecular Sciences.

  1. In section 6.2, line 401, “cytologic granules”, should be modified to “cytotoxic molecules”.

Thanks for your attention.

Author Response

In section 6.2, line 401, “cytologic granules”, should be modified to “cytotoxic molecules”.

Response: Thank you. We revised the text.